# Characterization of the highly fractured zone at the Grimsel test site based on hydraulic tomography

Lisa Maria Ringel[1], Mohammadreza Jalali[2], and Peter Bayer[1]

[1]Applied Geology, Institute of Geosciences and Geography, MLU Halle-Wittenberg, Halle, Germany
[2]Department of Engineering Geology and Hydrogeology, RWTH Aachen, Aachen, Germany

**Correspondence:** Lisa Maria Ringel (lisa.ringel@geo.uni-halle.de)

**Abstract.** In this study, we infer the structural and hydraulic properties of the highly fractured zone at the Grimsel test site in Switzerland by a stochastic inversion method. The fractured rock is modeled directly as a discrete fracture network (DFN) within an impermeable rock matrix. Cross-hole transient pressure signals recorded from constant rate injection tests in different intervals provide the basis for the herein presented first field application of the inversion. The experimental setup is realized by a multi-packer system. The geological mapping of the structures intercepted by boreholes and data from previous studies that were undertaken as part of the in-situ stimulation and circulation (ISC) experiments facilitate the setup of the site-dependent conceptual and forward model. The inversion results show that two preferential flow paths between the two boreholes can be distinguished. One is dominated by fractures with large hydraulic apertures while the other path consists mainly of fractures with a smaller aperture. The probability of fractures linking both flow paths increases the closer we are at the second injection borehole. The results accord with the findings from other studies conducted at the site during the ISC measurement campaign and add new insights about the highly fractured zone at the prominent study site.

## 1 Introduction

Solid rocks such as in crystalline and bedrock formations typically have a compact matrix of low permeability. Water pathways are focused on mechanical discontinuities that separate individual rock blocks over multiple scales. Such fractures are commonly described as planar structures, which form a network that is hard to resolve at field sites. This is due to the high diversity and complexity of natural fracture networks, the difficulty to identify fracture connectivities and thus to interpret the hydraulic regime of an entire formation based on local fracture detection. Accordingly, fractured aquifer characterization represents a challenge, with relatively high costs for applying specialized field investigation techniques and for gathering a sufficient data set for reliable hydraulic description. The general poor understanding of how groundwater flows in fractured field sites is in contrast to the relevance of fractured environments that host elementary freshwater reservoirs worldwide (Chandra et al., 2019; Wilske et al., 2020; Spencer et al., 2021). Aside from this, adequate characterization of the properties of fractured field sites concerns many subsurface engineering applications, such as the planning and operation of enhanced geothermal systems (Vogler et al., 2017; Kittilä et al., 2020), the evaluation of potential sites for a nuclear waste repository (Follin et al., 2014; Li

et al., 2022), or the description of an excavation-induced damaged zone around tunnels and openings (Armand et al., 2014; de La Vaissière et al., 2015).

Depending on the chosen experimental setting and the available data, different interpretations of the hydraulic and structural properties of a fracture network are possible. A fractured site can be inspected locally by borehole data, e.g., core mapping and geophysical image logs such as optical or acoustic televiewer. The depth and orientation of structures intercepted by boreholes characterizes fracture intensity and prevalent fracture orientations (Armand et al., 2014; Krietsch et al., 2018; Chandra et al., 2019; Tan et al., 2020; Yin and Chen, 2020; Pavičić et al., 2021), and by fitting probability distributions to the parameters a statistical analysis can be conducted (Barthélémy et al., 2009; Massiot et al., 2017). Single-hole and cross-hole flow and tracer tests are employed to infer permeability and connectivity between different borehole intervals (Le Borgne et al., 2006; Follin et al., 2014; de La Vaissière et al., 2015; de La Bernardie et al., 2018; Jalali et al., 2018; Brixel et al., 2020b, a; Tan et al., 2020; Li et al., 2022), the velocity distribution (Kang et al., 2015), or transport properties (Kittilä et al., 2019; Lee et al., 2019).

Detailed insight about the properties of flow paths between adjacent boreholes can be gained by tomographic methods. The principle of all tomographic methods is perturbing the investigated system e.g., by an injection of fluid, a tracer, a thermal anomaly, or an electric current, and recording the response at nearby receivers. In particular, geophysical tomographic methods are applied for the characterization of the rock properties, the identification of fractured, in particular highly fractured zones, and for the monitoring of flow pathways (Deparis et al., 2008; Dorn et al., 2012; Robinson et al., 2016; Doetsch et al., 2020). This is frequently done in combination with hydrogeological methods (Day-Lewis et al., 2003; Chen et al., 2006; Dorn et al., 2013; Voorn et al., 2015; Giertzuch et al., 2021b, a). A comprehensive portrayal of geophysical methods for the investigation of fractured field sites and the potential target applications is given in Day-Lewis et al. (2017).

In contrast to geophysical exploration techniques, hydraulic, pneumatic or tracer tomography is based on a fluid or tracer injection at a source well. The response is recorded at different adjacent boreholes at different depth intervals. In most cases, the pressure signals or tracer arrival curves are evaluated by a continuous hydraulic conductivity distribution based on an equivalent porous media (EPM) concept (Yeh and Liu, 2000; Illman et al., 2008, 2009; Sharmeen et al., 2012; Zha et al., 2015, 2016; Zhao and Illman, 2017; Dong et al., 2019; Zhao et al., 2019; Kittilä et al., 2020; Tiedeman and Barrash, 2020; Poduri et al., 2021; Zhao et al., 2021; Jiang et al., 2022; Liu et al., 2022). Thereby, detected high conductivity zones correspond with the locations of fractures or faults. Further insights about the fracture properties and improved results can be gained by particle tracking simulations (Tiedeman and Barrash, 2020), binary priors representing either fracture or matrix (Poduri et al., 2021), or by generating synthetic models with similar features like the field site (Zha et al., 2015). Geostatistical methods apply a stochastic EPM and different realizations of the subsurface are evaluated (Park et al., 2004; Blessent et al., 2011; Wang et al., 2017). Here, different facies represent different levels of fractured or intact rock, for which hydraulic conductivities are calibrated. In contrast to the EPM approach, the properties of the fracture network are inferred more directly by calibrating a connectivity pattern (Fischer et al., 2018b, a; Klepikova et al., 2020).

Our inversion approach differs from previous studies insofar as the fractured rock is represented explicitly as a discrete fracture network (DFN) and the hydraulic and structural parameters of the fractures are inferred directly. The great number of unknown parameters prevents the minimization of an objective function between simulated and observed data resulting in

a single deterministic DFN. Instead, a stochastic approach is applied to consider the non-uniqueness of the results. This is accomplished by generating several realizations of the fracture network that are equally likely to be evaluated as a fracture probability map. The validity of the approach was demonstrated for synthetic test cases in two dimensions (2D) (Somogyvári et al., 2017; Ringel et al., 2019) and three dimensions (3D) (Ringel et al., 2021). In this study, the new inversion method is applied to field data for the first time. We use transient pressure signals from hydraulic tomography experiments conducted as part of the in-situ stimulation and circulation (ISC) experiments at the Grimsel test site in Switzerland (GTS). Proper evaluation and validation of a new approach requires controlled tests and the GTS and ISC experiments pose a well-explored site for experimental validation. The objective of this paper is to reveal the feasibility and capability of 3D DFN inversion with a small-scale example. This study provides an elementary link between the theoretical development of a new inversion algorithm based on synthetic test cases and field applications although the small scale may not be representative of the much larger scale of groundwater reservoirs.

The paper is structured as follows: In the first part, we describe the site and the hydraulic tomography experiments to be used for the inversion. The implementation of the inversion is elaborated in the second part. We review the forward modeling procedure and the general inversion framework developed in previous works with synthetic test cases. We then explain the site-dependent inversion setting, i.e., the conceptual model and the prior parameter distributions that serve as basis for a stochastic inversion procedure and discuss and justify the necessary constraints and assumptions. The inversion results are interpreted and compared with findings from related ISC experiments.

## 2 Experimental setting

### 2.1 Test site

The GTS is an underground rock laboratory located in the Aar Massif in the Swiss Alps. The ISC experiments that serve as basis for this study utilized 15 boreholes of $20\,\mathrm{m}$ to $50\,\mathrm{m}$ depth, including two injection boreholes (INJ1 and INJ2). The other boreholes are used for stress and strain measurement, seismic, pressure, and temperature monitoring during the hydraulic stimulation phases (Krietsch et al., 2018). A general overview of the site with the persistent structures and the boreholes is shown in Fig. 1a. A summary of the experiments conducted during the ISC measurement campaign and their results are given in Amann et al. (2018) and Doetsch et al. (2018).

The crystalline rock at the southern part of the GTS (ISC experiment volume) has been moderately fractured. Ductile (S1) and brittle-ductile (S3) shear zones can be distinguished through the investigated rock volume (Fig. 1a) (Krietsch et al., 2018). The shear zones consist of a fault core, a damage zone and an unperturbed host rock (Wenning et al., 2018). A $4\,\mathrm{m}$ to $6\,\mathrm{m}$ highly fractured zone with fracture density (P10) around $3\,\mathrm{m}^{-1}$ is present between the fault cores of the two S3 shear zones, which is displayed in Fig. 1b. The fractures can be distinguished in wall damage zones adjacent to the S3 faults and linking damage zones, i.e., fractures connecting both fault cores (Brixel et al., 2020b). Testing campaigns on the connectivity between several intervals of the injection boreholes revealed that the best response occurs between the intervals 3 and 4 of both injection boreholes, which are located in the aforementioned highly fractured zone. Therefore, this is not only a highly fractured zone

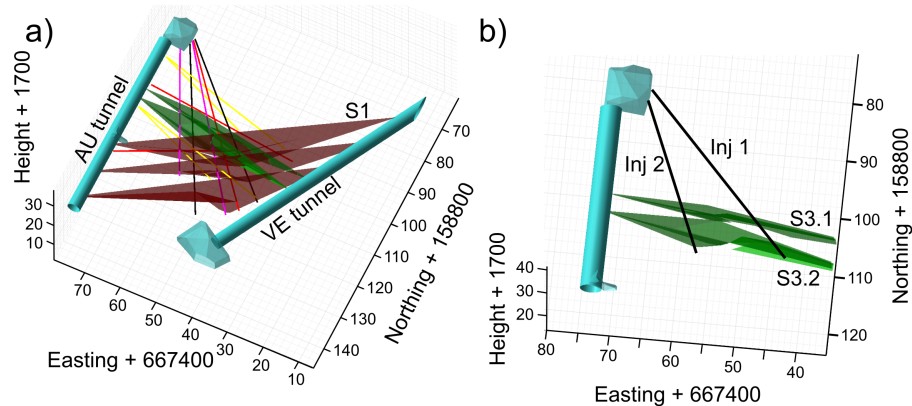

**Figure 1.** a) General overview of the ISC experiments site with the tunnels, all boreholes, and the two types of shear zones (Krietsch et al., 2018) and b) the volume that is investigated in this study, i.e., the zone between the two S3 faults.

**Table 1.** Parameters of the packer intervals and the hydraulic tomography experiments.

| Interval | Interval depth [m] | Injection flow rate [ml min$^{-1}$] | Injection time [min] |
|----------|--------------------|-------------------------------------|----------------------|
| Inj1-Int3 | $30 - 34$ | 60 | 60 |
| Inj1-Int4 | $27 - 29$ | 400 | 30 |
| Inj2-Int3 | $25 - 29$ | 60 | 60 |
| Inj2-Int4 | $22 - 24$ | 400 | 12 |

but also the most permeable region with conductive fractures (Jalali et al., 2018). For this reason, the characterization of the hydraulic and structural properties of this region (Fig. 1b) is the target of this study. The geological mapping of the structures intercepted by the boreholes and tunnels provides the basis for the setup of the conceptual model (Krietsch et al., 2018).

## 2.2 Hydraulic tomography data

The hydraulic tomography tests that are applied in this study are part of the characterization phase of the ISC experiment. We utilize transient pressure signals from constant rate injection tests in the intervals 3 and 4 of the injection boreholes INJ1 and INJ2. The different intervals are isolated by a multi-packer system. The properties of the packer intervals and the parameters of the injection are summarized in Table 1. Between each injection experiment, pressure recovery was possible. The pressure response of the fluid is measured using piezoresistive pressure transducers. The resolution of the pressure response data is in the range of $0.5\,\mathrm{kPa}$. The minimum principal stress is in the order of $8\,\mathrm{MPa}$. Since the injected fluid pressure is much below the minimum principal stress, the coupling between hydraulic and mechanical effects can be neglected in the forward modeling of

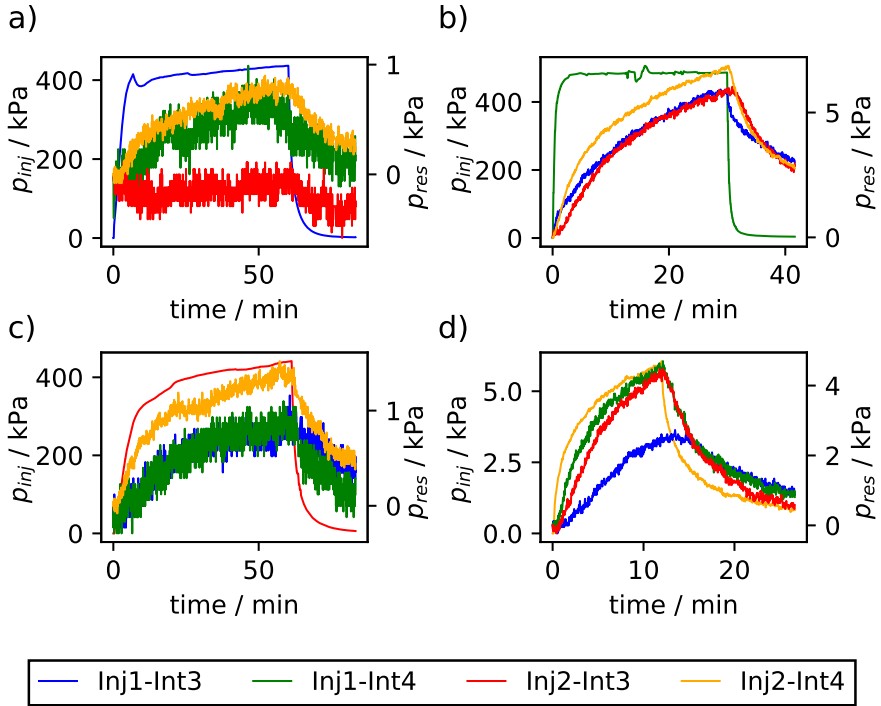

**Figure 2.** Pressure response in the different intervals provoked by a constant rate injection applied sequentially to the intervals Inj1-Int3 (a), Inj1-Int4 (b), Inj2-Int3 (c), and Inj2-Int4 (d) according to Table 1. The pressure measured in the respective injection interval belongs to the left vertical axes and the pressure signals measured in the observation intervals to the right vertical axes.

the experiment. The fluid pressure is measured with $\Delta t = 2\,\mathrm{s}$. In general, we use similar hydraulic tomography experiments as applied by Klepikova et al. (2020) except for a shorter injection time (Table 1) for computational reasons. The pressure signals are shown in Fig. 2 for each injection interval. Due to the stochastic inversion approach, the noisy pressure response data can be applied directly for the inversion without the necessity of smoothing or filtering the signals.

## 3 Implementation of the inversion

### 3.1 Forward modeling

Fractures are modeled as 2D objects with constant properties normal to the fracture midplane in a 3D rock matrix that is assumed impermeable. The pressure diffusion in a single fracture is described by

$$a\rho S\frac{\partial p}{\partial t} - \nabla_T \cdot \left(a\rho\frac{k_f}{\mu}\nabla_T p\right) = aq \tag{1}$$

with the hydraulic aperture $a$ [m], the density of the fluid $\rho$ [$\mathrm{kg\,m^{-3}}$], the specific storage $S$ [$\mathrm{Pa^{-1}}$], the permeability $k_f$ [$\mathrm{m^2}$], the dynamic viscosity $\mu$ [$\mathrm{Pa\,s}$], and a source/sink term $q$ [$\mathrm{kg\,m^{-3}\,s^{-1}}$]. The pressure $p$ [Pa] consists of the static pressure and

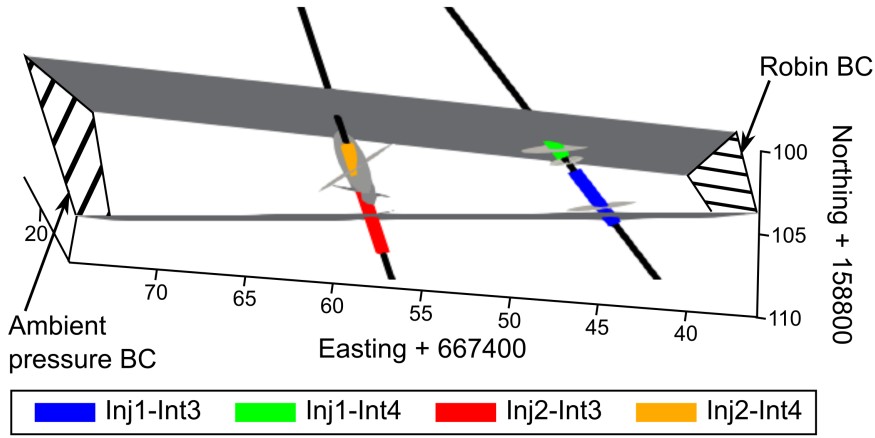

**Figure 3.** Overview of the volume considered in the forward model and the boundary conditions (BC). The geometry of the S3 faults is simplified to planes, the fractures intercepted by the injection intervals are illustrated as plane ellipses.

the piezometric pressure. The permeability is related to the aperture by

$$k_f = \frac{a^2}{12} \tag{2}$$

and the subscript T of the gradient ($\nabla_T$) denotes that it is evaluated in the fracture plane (Zimmerman and Bodvarsson, 1996; Berre et al., 2019). In this study, flow in the shear zones is modeled with the same approach as flow in the DFN, i.e., the shear zones are represented as 2D objects whereby the flow parameters are given by hydraulic aperture and specific storage (Eq. 1). The equations are solved numerically by the finite element method (FEM) with a conforming discretization at the intersections

of different fractures. The generation of the geometry and the meshing of the fractures and shear zones are implemented using the open-source mesh generator *Gmsh* (Geuzaine and Remacle, 2009). The geometry of each structure is created separately by the built-in geometry module of *Gmsh*. The fractures and the shear zones are connected for a conforming discretization at the intersections of different structures by the Boolean operations implemented in *Gmsh*.

The implemented boundary conditions are shown in Fig. 3 together with the S3 faults, and the fractures intercepted by the

injection boreholes obtained from optical televiewer logs (Krietsch et al., 2018). The boundary conditions are chosen under consideration that only a small volume of the ISC experiments is investigated in this study. Therefore, the following boundary conditions are applied: The AU tunnel is represented by a pressure boundary condition, in this case, ambient pressure. The way to the VE tunnel cannot be modeled explicitly. Therefore, we apply a Robin boundary condition as a transfer boundary condition to consider the transition of the flow and the extension of the shear zones towards the VE tunnel (Watanabe et al.,

2017). A no-flow boundary condition is applied normal to the planes of the fractures and shear zones.

## 3.2 Inversion algorithm

The parameters of the DFN $\theta$ are treated as unknowns characterized by probability density functions. Based on the Bayesian equation, the posterior density function $p(\theta|\mathbf{d})$ of the parameters given the measured data $\mathbf{d}$ is proportional to the likelihood function

$$\log L(\mathbf{d}|\theta) \propto -\sum_{i=1}^{N_{\text{data}}} \frac{(d_i - f(\theta)_i)^2}{2\sigma_i^2} \tag{3}$$

and the prior distributions $p(\theta)$ (Gelman et al., 2013). The term $f(\theta)$ refers to the forward simulation of the hydraulic tomography experiment for the DFN realization defined by the parameters $\theta$. The posterior density function is evaluated by sampling from it according to Markov chain Monte Carlo (MCMC) methods. This is an iterative procedure whereby new samples $\theta'$ are proposed by a proposal function and accepted ($\theta_i = \theta'$) with probability

$$\alpha = \min\left(1, \frac{p(\theta'|\mathbf{d})q(\theta_{i-1}|\theta')}{p(\theta_{i-1}|\mathbf{d})q(\theta'|\theta_{i-1})}|J|\right), \tag{4}$$

or rejected ($\theta_i = \theta_{i-1}$) (Brooks et al., 2011). The so-called reversible jump MCMC algorithm allows to change the number of parameters (Green, 1995). In this study, the number of parameters is adjusted by deleting or inserting a fracture within the prior bounds. The determinant of the Jacobian matrix $|J|$ has to be considered for transdimensional updates. It equals one for parameters sampled from the prior without linking its value to already existing parameters (Sambridge et al., 2006). The parameters of the inversion problem are adjusted by proposing a new value from a normal distribution whereby the mean of the normal distribution is given by the current value.

The variance $\sigma^2$ in the likelihood function (Eq. 3) accounts for different sources of uncertainties like measurement errors, modeling errors, and errors of the conceptual model. Therefore, the value of the variance is estimated separately for each pressure signal. This is implemented as part of the inversion algorithm after the update of the parameters of the DFN. The measured data is assumed to consist of a mean and a normally distributed error $\mathbf{d} = \overline{\mathbf{d}} + \mathcal{N}(0, \sigma^2)$. With this assumption, the variance can be estimated by sampling from an inverse gamma distribution

$$\sigma^2|\mathbf{d}, \theta \sim \mathcal{IG}\left(\frac{N_{\text{data}}}{2}, \frac{\sum_{i=1}^{N_{\text{data}}}(d_i - f(\theta)_i)^2}{2}\right) \tag{5}$$

as introduced by Gelman (2006) and implemented by e.g., Haario et al. (2006) and Ringel et al. (2019). For this reason, the noisy measured data can be applied directly for the inversion without filtering or smoothing the signals.

In practice, one iteration of the inversion algorithm operates as follows: Assuming the insertion of a fracture is chosen in the MCMC algorithm, the parameters (position, length, fracture set, i.e., orientation, hydraulic aperture) of the fracture are generated from the prior functions. The chosen parameters are evaluated by simulating the hydraulic tomography experiment with the proposed parameter set $\theta$, i.e., including the new fracture. The outcome of the simulation is compared to the measured pressure signals. If the error is smaller (the likelihood, Eq. 3, is higher) or similar to the previous step (without the fracture), the acceptance probability (Eq. 4) is high (Ringel et al., 2021). After accepting or rejecting the proposed parameters, the variance is updated according to Eq. 5.

## 3.3 Inversion constraints

The overall inversion procedure relies on several simplifications concerning parameters with less importance for our research target. For instance, the parameters specifying the properties of the shear zones have to be fixed. In general, our aim is an optimal balance between the accuracy of the generated results and the computational costs of the inversion procedure.

The underlying conceptual model comprises simplifications of the properties of single fractures that serve as inversion constraints. We assume plane ellipses as the fracture shape and the length of the minor axis equals half of the length of the major axis, i.e., the length ratio is fixed. The assumption of reducing the fracture shape to a 2D plane is a common assumption and justified by the derivation of the cubic law and the large ratio between the fracture extensions and the fracture aperture (Zimmerman and Bodvarsson, 1996). The assumption of the fracture shape as ellipse is reasonable since the flow is dominated by the path between the intersections of different fractures. Therefore, no sharp edges are considered for the simulation of the flow in the DFN. The hydraulic aperture is assumed constant over the fracture plane. Two fracture sets are defined with fixed orientations based on the orientations of the structures intercepted by the two injection boreholes. Thereby, the fracture set is chosen by the inversion algorithm for the fractures between the boreholes, however, the orientation assigned to the fracture sets are default. Figure 4 shows the orientation of the structures between the S3 shear zones intercepted by the two injection boreholes and the orientations defined for the two fracture sets. The appearance and distribution of the fractures dominate the flow. Accordingly, the surrounding rock matrix is assumed impermeable.

The investigated volume is limited to the volume between the two S3 shear zones (Fig. 1). The shear zones consist of a fault core and a damage zone. The permeability increases with the distance to the fault core whereby the cores are almost impermeable (Wenning et al., 2018). Since the properties of the shear zones are not the target of this study, the shape is simplified and the associated hydraulic parameters are fixed. The shape of the shear zones is simplified to a plane rectangle, i.e., a linear interpolation between the shear zones' traces at the injection boreholes. A constant hydraulic aperture of $a_{SZ} = 1 \cdot 10^{-5}\,\mathrm{m}$ is assigned. This small value is chosen based on preliminary in-situ tests and the knowledge, that the cores of the shear zones are impermeable at their tunnel intersection. A higher permeability of the shear zone at specific locations can be covered by placing fractures in the respective area which accounts also for the spatial variability of the permeability of the shear zone. The specific storage value is also fixed to $S_{SZ} = 1 \cdot 10^{-5}\,\mathrm{Pa}^{-1}$. The high value is prescribed considering the results from cross-borehole tests (Klepikova et al., 2020). Fractures of the fracture set 1 are approximately parallel to the S3 faults. Hence, a position close to an S3 fault also accounts for spatial changes in the permeability and specific storage of the S3 faults. Overall, the application of constraints and assumptions about the fracture shape limit an exact reproduction of the structural properties of the tested rock mass. However, those parameters that have a major influence on the flow in the DFN are adjusted by the inversion algorithm within prescribed bounds. These are in particular the position and the hydraulic aperture of fractures. In contrast, parameters with minor effects on the flow behavior are fixed, e.g., the exact fracture orientation or the length ratio.

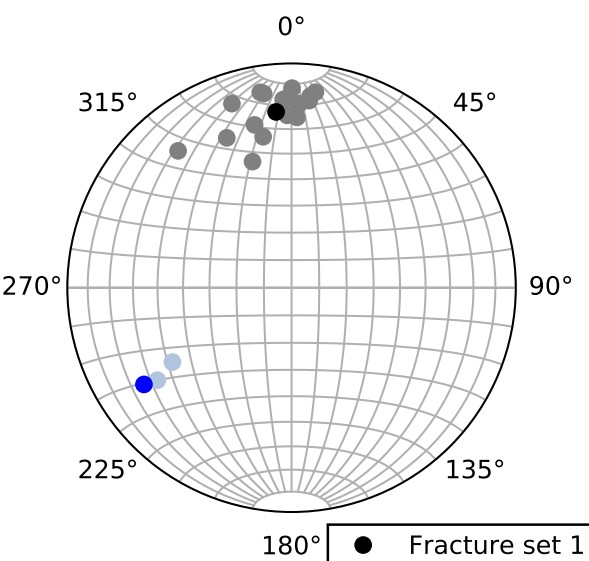

**Figure 4.** Orientations of the structures between the fault cores of the S3 shear zones in the injection boreholes observed from optical televiewer logs (Krietsch et al., 2018) in gray and light blue together with the calculated orientations for the fracture sets applied for the conceptual model.

### 3.4 Prior distributions

The parameters to be inferred are the number of fractures, the position of the fractures, the fracture lengths, the hydraulic aperture separately for each fracture, and the specific storage coefficient that applies to the whole DFN. The specific storage $S$ (Eq. 1) is given by the compressibility of water in theory (Freeze and Cherry, 1979). However, some fractures are only partially open and due to the roughness of the surface, the specific storage can be increased compared to the theoretical value (Jalali et al., 2018). Moreover, the hydraulic aperture is in general smaller than the actual aperture (Berre et al., 2019). The specific storage is assumed to be valid for the whole DFN since two variable hydraulic parameters for each fracture are not feasible for the inversion algorithm. Accordingly, five different update types are implemented to be applied sequentially: the transdimensional update changes the number of parameters by either inserting a new fracture or deleting a fracture. The other update types keep the number of parameters constant but adjust position, length, hydraulic aperture, or the specific storage. For the update of the position, length, and hydraulic aperture, one fracture is chosen randomly and a new value is proposed by a random perturbation of the current value.

Uniform prior distributions are applied, i.e., a parameter is specified by a constant probability between minimum and maximum possible values that are given in Table 2. The spatial priors are derived in general from the position of fractures intersecting the injection boreholes. The maximum value in the $x$ direction corresponds to the distance to the AU tunnel to apply

**Table 2.** Uniform prior distributions defined by a minimum and maximum possible value.

|  | Minimum | Maximum |
|---|---|---|
| $x$ (Easting + 667,400) / m | 45 | 70 |
| $y$ (Northing + 158,800) / m | 102 | 108 |
| $z$ (Height + 1,700) / m | 14 | 19 |
| Fracture length / m | 0.4 | 7 |
| Hydraulic aperture / m | $1 \cdot 10^{-5}$ | $1 \cdot 10^{-3}$ |
| Specific storage / $\mathrm{Pa}^{-1}$ | $5 \cdot 10^{-10}$ | $1 \cdot 10^{-6}$ |

All spatial values refer to the position of the midpoint of the ellipse.

the boundary condition. The prior for the north direction is given such that the fractures are located between the cores of the S3 shear zones. The elevation of fractures is expected to have a minor influence on the flow between the two boreholes, and a broader possible range for the elevation would be less resolved. In the following, $x$ refers to Easting $+ 667,400\,\mathrm{m}$, $y$ to Northing $+ 158,800\,\mathrm{m}$, and $z$ to Height $+ 1,700\,\mathrm{m}$ (Fig. 1). The minimum value for the fracture length is given by the borehole diameter, and the maximum possible value corresponds with the distance between the shear zones. Fractures proposed during iterative inversion which intersect with the fault cores of the shear zones are reduced to the part of the fracture within the investigated volume (Fig. 1b and Fig. 3).

The prior range for the aperture is approximated from the results of single- and cross-borehole tests (Jalali et al., 2018; Brixel et al., 2020b, a). The minimum specific storage value is given by the compressibility of water (Freeze and Cherry, 1979), whereas the maximum value is based on cross-borehole injection tests (Klepikova et al., 2020). Both prior distributions for the hydraulic parameters cause the flow preferentially in the DFN rather than in the shear zones, due to a smaller specific storage and a larger hydraulic aperture of the fractures.

## 4 Results

### 4.1 Processing of the results

Overall, 27,000 DFN realizations are considered as posterior DFN realizations since they minimize the error and fulfill the prior conditions. DFN realizations from the initial 500 iterations are discarded as so-called burn-in iterations due to a higher error. The computation of the inversion was executed by an Intel Core i9 Workstation with 10 Cores and $128\,\mathrm{GB}$ RAM and lasted about one week. The posterior realizations are approximately equally likely. They reflect the uncertainty of the inversion results in contrast to a unique solution that would be obtained by a deterministic approach. To reduce the autocorrelation of the results, we keep every 100th realization for further processing, which is called thinning (Brooks et al., 2011). By the stochastic

approach applied here, the fit between the measured and simulated pressure signals of the hydraulic tomography experiment is evaluated by the posterior and prediction uncertainty. The posterior limits are calculated based on the simulated pressure signals of the posterior DFN realizations which corresponds with the uncertainty of the inversion method. The uncertainty for predicting new observations is a measure for the overall error, as well as conceptual simplifications, since it also considers the estimated variance (Eq. 5). The DFN realizations are evaluated by a fracture probability map (FPM) over the investigated volume. For this, the inspected rock volume is divided into raster elements. Each element records if the element is part of a fracture. By taking the mean element-wise over all the posterior DFN realizations, the probability of each raster element for being part of a fracture is derived. The evaluation of the FPM summarizes the estimated position and length of the fractures, i.e., those parameters with major influence on the flow. The hydraulic aperture is evaluated on the same raster elements. If a raster element is part of a DFN realization, the respective aperture is taken from the DFN. Thereby, the mean hydraulic aperture can be evaluated for each element.

## 4.2 Evaluation of the data

In the first step, the measured and simulated pressure signals are compared to assess the quality of the posterior realizations. Figure 5 shows the median fit and the $95\%$ limits of the forward simulation of the posterior DFN realizations and the $95\%$ limits of the prediction uncertainty together with the observed data. Figure 5 demonstrates that the general shape and trend of the measured signals are reproduced by the simulated pressure curves checking the median fit and the $95\%$ posterior limits. This is the case especially for the response in the intervals 4 of both boreholes. The weaker fit of some signals in the intervals 3 indicates effects not covered by the inversion approach or forward simulations, such as deviations from the assumed fracture shape or fracture orientations. For a given DFN realization, the actual measured pressure signals are predicted. Due to measurement noise and simplifications concerning the DFN model, the $95\%$ limits of the prediction uncertainty are wider than for the posterior uncertainty.

## 4.3 Evaluation of the DFN realizations

The FPM and the mean hydraulic aperture are shown for different cross sections $z$ in Fig. 6. The fractures intercepted by the injection intervals and the shear zones are fixed, and therefore, they appear with a probability of $100\%$. Their orientation as derived from the optical televiewer logs is assigned to these fractures, therefore, the orientation is in the same range as the orientation defined for the fracture set, but the exact values vary. Overall, two different connections with different levels of permeability are present. A flow path dominated by fractures with a large hydraulic aperture exists between the injection intervals 4 of both boreholes (Inj1-Int4 and Inj2-Int4). The fractures providing this connection are visible with a high probability in the cross sections $z = 16\,\mathrm{m}$ and mainly $z = 17\,\mathrm{m}$. In general, a good respectively permeable connection between two intervals is possible by a large hydraulic aperture of the fractures, long fractures, by a long intersection length between different fractures, or by a correlation of these factors. In contrast, a connection with fractures with smaller hydraulic apertures appears between both intervals 3 (Inj1-Int3 and Inj2-Int3) and Inj2-Int4. This flow path is present with an average probability of approximately $50\%$ primarily in the cross section $z = 15\,\mathrm{m}$. Fractures linking both flow paths appear more likely the closer the location is to

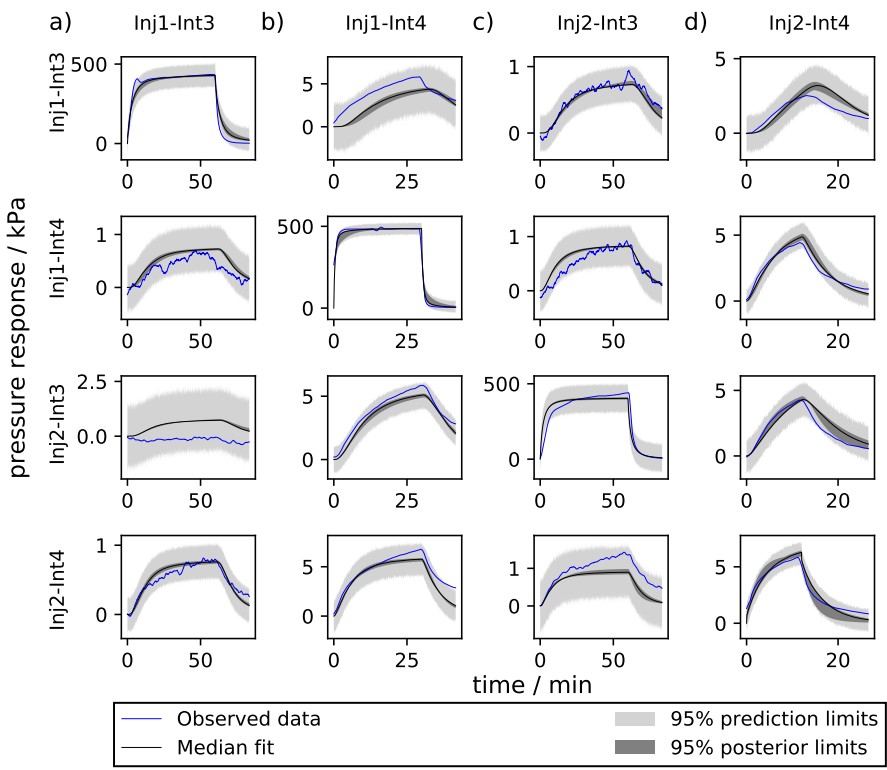

**Figure 5.** Comparison of the observed pressure response with the simulation of the hydraulic tomography experiment for the posterior DFN realizations for the injection in the intervals Inj1-Int3 (a), Inj1-Int4 (b), Inj2-Int3 (c), and Inj2-Int4 (d) according to Table 1. For the clarity of the visualization, the observed pressure signals are smoothed.

injection borehole 2, i.e., further east. The described behavior is also reflected in the measured data. All responses provoked by the injection in both intervals 4 are more distinct than for the injection in the intervals 3. Although a maximum hydraulic aperture of $10^{-3}$ m is enabled by the prior distribution, only a few fractures with a small probability appear with an aperture close to the maximum possible value as visible in Fig. 6 at a depth of $z = 17$ m. The specific storage coefficient converges

to a mean value of $S = 7.4 \cdot 10^{-7} \, \mathrm{Pa}^{-1}$. Only a few updates were possible that occurred mainly during the burn-in iterations. Therefore, this value is interpreted as the result of an optimization, i.e., as the averaged specific storage to be applied for the whole DFN. The estimated specific storage is greater than the theoretical value that functioned as the minimum value of the prior distribution of the specific storage (Tab. 2). This considers a delay in the response that is not related exclusively to the compressibility of water (Freeze and Cherry, 1979) but also to e.g., the surface roughness or fractures that are only partially

open. Multiplied with the maximum possible aperture (Tab. 2), the inferred value is well within the storativity range calculated from cross-borehole injection tests (Klepikova et al., 2020). Several fractures of fracture set 1 appear close to the S3.1 shear zone indicating either permeable fractures close to the shear zone or a higher permeability of the shear zone in this region than the assigned value. This demonstrates that the prescribed assumptions on hydraulic properties of the shear zone do not induce

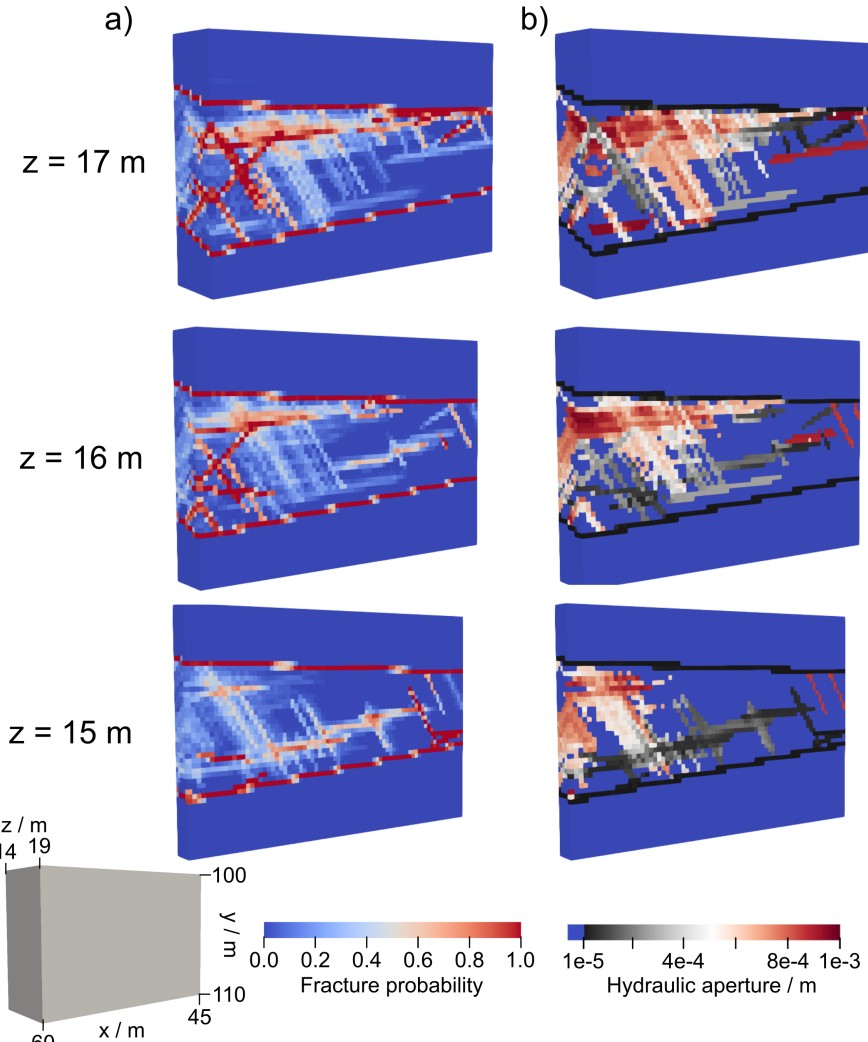

**Figure 6.** Evaluation of the results by the fracture probability map (a) and the mean hydraulic aperture (b) for different cross sections $z$. The boundaries of the investigated volume are indicated by the cuboid in the lower left.

crucial conceptual constraints in the inversion, but a locally high permeability of a shear zone is indicated by a locally high fracture probability.

Although the volume east from injection borehole 2 towards the AU tunnel is part of the inversion, i.e., fractures can be inserted or moved in this volume, the resolution of the results is low since various DFN realizations, i.e., fracture positions, are possible. Only the volume between the two injection boreholes can be evaluated with a sufficient resolution.

## 4.4 Comparison with other studies

The inferred flow paths consist of fractures with a high or rather low permeability which accords with the results from Klepikova et al. (2020). We also compare our results with the structures intersected by other boreholes drilled after conducting the experiments evaluated in this study. While this inversion approach is capable of identifying fractures that are hydraulically relevant, geophysical methods, such as optical televiewer logs report all structures intercepted by boreholes independent of their permeability. The boreholes PRP1 and FBS1 are partially located within the prior range defined for this study. The interval

at a depth of $23\,\mathrm{m}$ to $25\,\mathrm{m}$ of PRP1 has been identified as the interval with the highest transmissivity by Kittilä et al. (2019) and Brixel et al. (2020a). In $95\%$ of the posterior DFN realizations, at least one fracture is present in this interval. Fractures that intersect with the interval between the S3 faults of the FBS1 borehole are present in about $45\%$ of the posterior DFN realizations. This supports that crucial hydraulic features of the DFN can be identified by the presented inversion approach. Still, even if such successful local validation is possible, there are no other independent measurements available to confirm the

validity of the inverted complete DFN structure and its probability. Geophysical measurements such as seismic data (Doetsch et al., 2020) or ground-penetrating radar (Giertzuch et al., 2021a) were able to characterize the ISC volume on a decameter-scale and identify the persistent structures and the highly fractured zone, however, they could not delineate or specify the properties of single flow paths.

## 5 Conclusions

In this study, we characterized the highly fractured zone at the GTS based on transient pressure signals from hydraulic tomography experiments with a new stochastic inversion method. A stochastic approach was applied to assess the uncertainty of the measured data and the non-uniqueness of the results. The fractured rock is represented directly as a DFN model in the forward simulations. Several posterior DFN realizations that are approximately equally likely are evaluated, and two preferential flow paths dominated by a large or small hydraulic aperture are successfully identified. The presented method relies on

some investigations to be applied prior to the inversion, such as the mapping of structures intercepted by boreholes, as well as it benefits from single- and cross-hole permeability tests for the definition of a range of the hydraulic aperture. In case it is possible to further narrow down the prior range of the hydraulic parameters, the specific storage can be inferred separately for each fracture instead of computing only a mean value for the whole DFN. In general, improved results and more insights about the fractured rock can be gained by the same inversion method but with more pressure signals from additional intervals and

boreholes.

Future research is necessary on the generally most suitable definition of prior and proposal distributions, which are elementary for robust inversion and for deriving meaningful results. The efficiency of the MCMC sampling can be improved significantly by implementing more elaborate prior or proposal distributions, for example, relying on soft information and site-specific expertise. A further option is utilizing continuous inversion results, such as continuous hydraulic conductivity dis-

310 tributions, or geophysical measurements for highlighting a priori regions with a higher probability for the insertion of fractures

or to define zones that are likely connected by fractures to reduce the number of necessary inversion iterations (Dong et al., 2019).

The introduced inversion framework can be applied in a highly flexible way for the characterization of different fractured sites by adapting the site-dependent parameters to meet the conditions of the tomography experiment at each site. Moreover, different types and sources of measured data can be processed for the inversion, such as tracer or in-situ stress data, provided that a forward model is available that allows for the flexible update of DFN parameters. The workflow for the setup of the inversion problem is similar. The basis is the properties of the fractures intercepted by the boreholes, i.e., their position and orientation, obtained from optical or acoustic televiewer logs or outcrops. This knowledge is utilized for the prior distributions on the spatial parameters and for the specification of fracture sets. The prior distributions on the hydraulic parameters are based on cross-hole flow tests in this study. This can also be done by the evaluation of the hydraulic tomography experiments as continuous hydraulic conductivity and specific storage tomogram. As the definition of priors and constraints delineates the range of feasible DFN realizations, this step has to be done carefully. However, the presented Bayesian framework allows the combination of multiple and diverse hard and soft data, which often exists in addition to hydraulic test data that is used to guide the inversion. As demonstrated here, too tight constraints may be avoided by uniform prior distributions with large value ranges at the expense of higher computational costs for the inversion. In practice, the amount of information describing the fractured rock is determined mainly by the hydraulic tomography data, i.e., by the number of intervals and boreholes.

The present study paves the way towards the applicability of the discrete inversion approach on a larger scale. The main issue will be to balance the degree of field testing with the desired fracture resolution and the associated computational costs. One possible direction is explicitly implementing only large conductive fractures. The role of smaller fractures with a lower permeability could be represented by calibrating a background permeability within the discrete fracture matrix approach (Berre et al., 2019). Another appealing direction is the representation of scale-dependent fracture sets by their statistical properties following a hierarchical parameterization (Ma et al., 2020).

*Data availability.* The geological data set used for the setup of the conceptual model is available from Krietsch et al. (2018). The measured pressure curves are available from Jalali et al. (2022).

*Author contributions.* Funding acquisition: P.B.; Measurements: M.J.; Methodology: L.M.R., P.B.; Inversion: L.M.R; Writing – original draft: L.M.R.; Writing – review & editing: M.J., P.B.

*Competing interests.* The authors declare no conflict of interest.

*Acknowledgements.* This research was funded by the German Research Foundation (DFG), grant number BA-2850-5-1. We thank Ryan Pearson for language editing and two anonymous referees for their valuable comments which helped to improve the quality of the manuscript.

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
