# Peer review of "Characterization of the highly fractured zone at the Grimsel test site based on hydraulic tomography"

_Hydrology and Earth System Sciences, 2022_

## Author Comment (AC3)

Correction to our reply to referee #2:

Referee comment: *Figure 6 – It is difficult to relate the z positions to the model volume (no z axis or magnitudes shown in the very small cuboid reference diagram). Also it is difficult to read aperture magnitudes with the increment labeling on the shading scale. The text refers to east positioning (L255), so direction labels are needed here, and also in Fig 1. Enlarging the reference diagram would be helpful.*

In our previous reply, we misunderstood your comment concerning the aperture "labeling on the shading scale". We interpreted your comment as recommendation to use another colormap. However, the labeling was in fact wrong. We generated the figures for different cross sections with the program Paraview. When we inserted them to Figure 6 of our paper we relabeled the colorbar to provide the figure (especially the numbers) in a high quality where we made the mistake.

Please find the corrected version of Figure 6 of our paper and the input as generated with Paraview below.

[Figure]

*Figure 1: Corrected version of Figure 6 of our paper.*

[Figure]

*Figure 2: Original figure generated with Paraview for z = 15m*

[Figure]

*Figure 3: Original figure generated with Paraview for z = 16m*

[Figure]

*Figure 4: Original figure generated with Paraview for z = 17m*

---

## Author Response (AR1)

**Reply to comments by editor and referees:**

Explanation on replies:

| Text | Comment from editor or referee |
|------|-------------------------------|
| Text | Comment from authors |
| Text | Original manuscript text |
| Text | New manuscript text |
|  |  |

Line numbers refer to the revised manuscript with the changes tracked.

**Editor comment:**

*Two reviews and author responses have been received.*

*One review requested minor revisions, and appropriate responses and proposed revisions have been provided by the authors. The other review is extensive and detailed, and raises many points that require serious thought and revision. In their response to this review, the authors provided a detailed series of arguments. Overall, the response appears to satisfactorily address the review. Furthermore, I believe a seriously revised (and appropriately expanded) manuscript can be considered as a regular article rather than a technical note.*

*I therefore recommend careful revision to the manuscript along the lines described in the authors' responses. Especially in the context of the long review, though, I encourage the authors to include revisions that address each and every point raised by the reviewer; this includes, e.g., a modification to the state "Objectives", and other comments that are addressed by the authors in their response, but which might not be accompanied by similar insertion of additional explanation in the revised manuscript itself. The reviewer raised many points that sharpen the contribution - and limitations - of the current work, and other points concerning clarity of methodology. Fully addressing all of these points in the revised manuscript, with additional explanations and clarifications (also of limitations), will lead to a stronger contribution.*

*I look forward to receiving the revised manuscript.*

> We revised the original version of the manuscript following the points raised by the referees. One concern is the limited number of intervals and the small scale of the problem. Therefore, we modified the objective of the paper as suggested to clarify that we chose a well-explored site and small-scale example as link between theoretical development and practical application of the new inversion algorithm. We also emphasize in the introduction (Line 62-75) and the conclusions of the paper (Line 335-340) that this is the first field application of a recently developed DFN inversion method and that further research and careful considerations are necessary before applying the method on a larger scale. Moreover, several comments were raised questioning the necessary assumptions/constraints/prior information and the actual contribution of the HT data. Therefore, we provide more arguments supporting the choice of inversion constraints (Line 170-199).

> Line 62-75:
> "In this study, the new inversion method is applied to field data for the first time. We use transient pressure signals from hydraulic tomography experiments conducted as part of the in-situ stimulation and circulation (ISC) experiments at the Grimsel test site in Switzerland (GTS). Proper evaluation and validation of a new approach requires controlled tests and the GTS and ISC experiments pose a well-explored site for experimental validation. The objective of this paper is to reveal the feasibility and capability of 3D DFN inversion with a small-scale example.  This study provides an elementary link between the theoretical development of a new inversion algorithm based on synthetic test cases and field applications although the small scale may not be representative of the much larger scale of groundwater reservoirs.

The paper is structured as follows: In the first part, we describe the site and the hydraulic tomography experiments to be used for the inversion. The implementation of the inversion is elaborated in the second part. We review the forward modeling procedure and the general inversion framework developed in previous works with synthetic test cases. We then explain the site-dependent inversion setting, i.e., the conceptual model and the prior parameter distributions that serve as basis for a stochastic inversion procedure and discuss and justify the necessary constraints and assumptions. The inversion results are interpreted and compared with findings from related ISC experiments."

**Comments by referee #1:**

*This exciting manuscript applies hydraulic tomography to delineate fractures in the geologic media. I would recommend a minor revision and publishing this manuscript. The followings are my suggestions for minor revisions.*

1. *Line 46. The manuscript should have reviewed the work by Yeh and Liu (2002), Illman et al, 2008; and Zha et al. (2016), and Dong et al. (2020), which applied hydraulic tomography to delineate fractures in geological media.*

We included the suggested references in the revised version of the manuscript.
Line 44-49:
"The response is recorded at different adjacent boreholes at different depth intervals. In most cases, the pressure signals or tracer arrival curves are evaluated by a continuous hydraulic conductivity distribution based on an equivalent porous media (EPM) concept (Yeh and Liu, 2000, Illman et al., 2008, Illman et al., 2009, Sharmeen et al., 2012, Zha et al., 2015, Zha et al., 2016, Zhao et al., 2017, Dong et al., 2019, Zhao et al., 2019, Kittilä et al., 2020, Tiedeman & Barrash, 2020, Poduri et al., 2021, Zhao et al., 2021, Jiang et al., 2022, Liu et al., 2022). Thereby, detected high conductivity zones correspond with the locations of fractures or faults."

2. *Line 120. You should have applied HT to equivalent porous media to find the likely connected fractures first as Dong et al. (2022) did. Afterward, generate DFN to fine-tune your HT results.*

An EPM was evaluated for the investigated region based on travel time tomography (Kittilä et al., 2020), however, the results are only 2D and not sufficient to estimate likely connected fractures. We added your suggestion to the conclusion to consider that for further research.
Line 316-319:
"A further option is utilizing continuous inversion results, such as continuous hydraulic conductivity transmissivity distributions, or geophysical measurements for highlighting a priori regions with a higher probability for the insertion of fractures or to define zones that are likely connected by fractures to reduce the number of necessary inversion iterations (Dong et al., 2019)."

*Dong, Y., Fu, Y., Yeh, T.-C. J., Wang, Y.-L., Zha, Y., Wang, L., & Hao, Y. (2019). Equivalence of discrete fracture network and porous media models by hydraulic tomography. Water Resources Research, 55. https://doi.org/10.1029/ 2018WR024290*

**Comments by referee #2:**

*General Comments:*
*This manuscript presents the first use of a stochastic inversion method for estimating the location and hydraulic parameters of a 3D discrete fracture network (DFN) using hydraulic tomography (HT) field testing. The field application is a volume centered on 2 shear zones at the well-studied Grimsel test site, and substantial use is made of data and experience from prior testing at this site. The objective of this paper is "to reveal the feasibility and capability of 3D DFN inversion in practice under well-explored*

*field conditions" (L65-66). Principal results are estimations of probable locations and parameterization of fracture connections between the 2 shear zones.*

We modified the description of the objective to clarify that this study links the theoretical development of the inversion algorithm based on synthetic test cases with the first field application at a well-explored site.

Line 62-75:

"In this study, the new inversion method is applied to field data for the first time. We use transient pressure signals from hydraulic tomography experiments conducted as part of the in-situ stimulation and circulation (ISC) experiments at the Grimsel test site in Switzerland (GTS). Proper evaluation and validation of a new approach requires controlled tests and the GTS and ISC experiments pose a well-explored site for experimental validation. Based on this application, tThe objective of this paper is to reveal the feasibility and capability of 3D DFN inversion with a small-scale example.in practice under well-explored field conditions This study provides an elementary link between the theoretical development of a new inversion algorithm based on synthetic test cases and field applications although the small scale may not be representative of the much larger scale of groundwater reservoirs.

The paper is structured as follows: In the first part, we describe the site and the hydraulic tomography experiments to be used for the inversion. The implementation of the inversion is elaborated in the second part. We review the forward modeling procedure and the general inversion framework developed in previous works with synthetic test cases. We then explain the site-dependent inversion setting, i.e., the conceptual model and the prior parameter distributions that serve as basis for a stochastic inversion procedure and discuss and justify the necessary constraints and assumptions. The inversion results are interpreted and compared with findings from related ISC experiments."

*My overall assessment of the manuscript is strongly affected by the very limited application of HT (only 4 tests between a total of only 4 packed-off intervals in 2 boreholes) in a very limited volume of investigation that is modeled and interpreted with the use of a large amount of prior characterization and testing information, simplifications, and assumptions (L166-207).*

First, we agree with the reviewer that we analyzed only a small rock volume, and there is considerable work and field testing needed in the future with greater rock volumes in order to establish the presented approach to support the management of groundwater and geothermal resources in practice. However, this is the first field application of the presented new inversion procedure. Proper testing and validation require well-controlled tomographic tests which are extremely rare and the Grimsel test site represents one of the most characterized fractured underground rock laboratories worldwide, which is a unique lab for experimental validation for decades. The small scale thus may not be representative of the much larger scale of groundwater reservoirs. Still, field laboratory validation—even only on a small scale—is considered an elementary link between theoretical development and full-scale demonstration in practice. We clarified that in the introduction (Line 62-75). Nevertheless, we agree that future research is necessary on the applicability of the inversion method on a larger scale and discuss this issue accordingly in the conclusions of the paper. We also discuss the general workflow and the necessary prior knowledge for applying the inversion method to other sites detailed in the conclusions of the revised manuscript.

Line 335-340:

"The present study paves the way towards the applicability of the discrete inversion approach on a larger scale. The main issue will be to balance the degree of field testing with the desired fracture resolution and the associated computational costs. One possible direction is explicitly implementing only large conductive fractures. The role of smaller fractures with a lower permeability could be represented by calibrating a background permeability within the discrete fracture matrix approach (Berre, 2019). Another appealing direction is the

representation of scale-dependent fracture sets by their statistical properties following a hierarchical parameterization (Ma et al., 2020)."

We carefully reported the extent of data and information used for characterization of the site. This may have caused the impression that the inversion is only feasible with an extraordinary amount of prior characterization and testing information, simplifications, and assumptions. Clearly, we did not provide a satisfactory argumentation in the original manuscript, and this was a shortcoming. As described in the following as response to specific comments, however, these premises are justified.

*In this regard also, I question whether the issue of non-uniqueness is sufficiently addressed – and perhaps only can be for this type of study with prediction of field validation tests in different multi-well, multi-zone testing configurations, and/or which might include tracer tests that have multi-zone multi-well monitoring.*

Addressing and revealing non-uniqueness is an elementary component of the presented inversion procedure. Note that fracture probabilities are derived in comparison to deterministic model configurations, such as common in equivalent porous media-based inversions. The non-uniqueness of the results is considered by applying a stochastic approach, by evaluating the fracture probability, and the posterior and prediction probability. Considering this, and the state-of-the art in this field, we are surprised that the reviewer questions the capability of the presented procedure. We tried to emphasize this more in the revised version:

Line 56-62:

"Our inversion approach differs from previous studies insofar as the fractured rock is represented explicitly as a discrete fracture network (DFN) and the hydraulic and structural parameters of the fractures are inferred directly. The great number of unknown parameters prevents the minimization of an objective function between simulated and observed data resulting in a single deterministic DFN.  Instead, a stochastic approach is applied to consider the non-uniqueness of the results. This is accomplished by generating several realizations of the fracture network that are equally likely to be evaluated as a fracture probability map. The validity of the approach was demonstrated for synthetic test cases in two dimensions (2D) (Somogyvári et al., 2017, Ringel et al., 2019) and three dimensions (3D) (Ringel et al., 2021)."

We agree with the reviewer that different multi-well, multi-zone, and tracer tests would be favorable. Unfortunately, such tests are not realistic for us due to cost and time reasons. In the conclusions we acknowledge that more injection intervals/boreholes could provide a better result.

Line 310-312:

"In general, improved results and more insights about the fractured rock can be gained by the same inversion method but with more pressure signals from additional intervals and boreholes."

Line 331-334:

"As demonstrated here, too tight constraints may be avoided by uniform prior distributions with a large value range at the expense of higher computational costs for the inversion. In practice, the amount of information describing the fractured rock is determined mainly by the hydraulic tomography data, i.e., by the number of intervals and boreholes."

*As currently presented, the combination of few HT tests with large amounts of prior data and assumptions (much of which restrict the range of possible fracture and fracture network occurrence - see Specific Comments below for L172-177 and what is shown in Fig 6 with particular relevance to this issue), and modeling the few tests within a small volume (i.e., close boundaries including 2 shear zones treated as having impermeable "cores"), raise questions about: (1) what the HT actually contributes to the results; (2) how sensitive the results are to the many individual and combined priors and*

*assumptions; and (3) how feasible and capable the method is for application at other sites that may not have such abundant prior information or small volume of investigation and modeling.*

(1): The HT data is the main factor for the acceptance criterion of the MCMC algorithm. For example, if the insertion of a fracture is chosen in the MCMC algorithm, the parameters (position, length, fracture set, i.e., orientation, hydraulic aperture) of the fracture are generated from the prior, respectively the proposal function (uniform prior or normal proposal function). The chosen parameters are evaluated by simulating the hydraulic tomography experiment with the new fracture and by comparing the simulated data with the measured data. If the error is smaller (the likelihood, Eq. 3, is higher) than in the previous step (without the fracture), the acceptance probability (Eq. 4) is high. Therefore, the comparison with the measured HT data is crucial for the inversion. The same procedure holds for e.g., the update of the hydraulic aperture or the position of a fracture.

(2): The mentioned prior information and simplifications/assumptions/constraints are justified and usually available also at other sites: one basis is the properties of fractures intersected by the boreholes. Their orientations and depths can be generally obtained easily from optical and/or acoustic televiewer logs or outcrops (Line 27—31). This is utilized in the presented approach as hard data to define the range of fracture positions. The prior distributions for the hydraulic properties can be defined from preliminary e.g., cross-hole flow tests or continuous conductivity and specific storage tomograms. In our study, we applied an even larger range for the hydraulic parameters than the range suggested by the cross-hole flow tests at our site to consider assumptions and errors from the evaluation of the flow tests. Despite the possibility of proposing a maximum aperture of $10^{-3}$ $m$, only few fractures with a small probability were estimated with this aperture since they cause an increased error in the evaluation of the likelihood function, i.e., this is rejected because of the HT data. The assumption of reducing the fracture shape to a 2D plane is a common assumption and justified by the derivation of the cubic law (Zimmerman and Bodvarsson, 1996). The assumption of the fracture shape as ellipse is reasonable because the flow is dominated by the path between the intersections of different fractures. Therefore, no sharp edges are considered for the simulation of the flow in the DFN. Restricting the orientation of fractures between the boreholes to only two different orientations is potentially a strong conceptual limitation. However, estimation of the exact fracture orientation by the inversion algorithm is much less important compared to inferring position, length, and hydraulic parameters. Prescribing fixed permeability values to the shear zones seems like a strong assumption. However, the inversion adjusts spatial variability of the permeability by placing fractures close to the S3 shear zones (see our answer to the next comments). In fact, the existence of nearby shear zones complicates the field validation in our application and represents an additional challenge that is overcome. Applying the same inversion framework to other sites without shear zones might be even easier.

(3): The necessary prior knowledge/assumptions/simplifications can be obtained in a similar strategy for different sites. We agree that further information and preliminary studies are necessary for the successful application of the DFN inversion method. We are certain that this is possible, since the most important values are the properties of fractures intersected by boreholes and an estimate of the hydraulic parameters from cross-hole flow tests.

The questions raised by the reviewer are considered in the revised manuscript with a more detailed explanation of the importance of the HT data in the acceptance probability.
Line 162-168:
"In practice, one iteration of the inversion algorithm operates as follows: Assuming the insertion of a fracture is chosen in the MCMC algorithm, the parameters (position, length, fracture set, i.e., orientation, hydraulic aperture) of the fracture are generated from the prior functions. The chosen parameters are evaluated by simulating the hydraulic tomography experiment with the proposed parameter set $\theta$, i.e., including the new fracture. The outcome

of the simulation is compared to the measured pressure signals. If the error is smaller (the likelihood, Eq. 3, is higher) or similar to the previous step (without the fracture), the acceptance probability (Eq. 4) is high (Ringel et al., 2021). After accepting or rejecting the proposed parameters, the variance is updated according to Eq. 5."

We also added a more detailed explanation of the definition of the inversion constraints and assumptions:
Line 173-179:
"The underlying conceptual model comprises simplifications of the properties of single fractures that serve as inversion constraints. We assume plane ellipses as the fracture shape and the length of the minor axis equals half of the length of the major axis, i.e., the length ratio is fixed. The assumption of reducing the fracture shape to a 2D plane is a common assumption and justified by the derivation of the cubic law and the large ratio between the fracture extensions and the fracture aperture (Zimmerman and Bodvarsson, 1996). The assumption of the fracture shape as ellipse is reasonable since the flow is dominated by the path between the intersections of different fractures. Therefore, no sharp edges are considered for the simulation of the flow in the DFN."
Line 195-199:
"Overall, the application of constraints and assumptions about the fracture shape limit an exact reproduction of the structural properties of the tested rock mass. However, those parameters that have a major influence on the flow in the DFN are adjusted by the inversion algorithm within prescribed bounds. These are in particular the position and the hydraulic aperture of fractures. In contrast, parameters with minor effects on the flow behavior are fixed, e.g., the exact fracture orientation or the length ratio."

We also explained how the prior distributions affect the results in Line 269-271:
"Although a maximum hydraulic aperture of $10^{-3}\ m$ is enabled by the prior distribution, only a few fractures with a small probability appear with an aperture close to the maximum possible value in Fig. 6 at a depth of z = 17 m."

Ideas and suggestions for transferring the approach to other sites are given in the conclusions.
Line 320-328:
"The introduced inversion framework can be applied in a highly flexible way for the characterization of different fractured sites by adapting the site-dependent parameters, mainly the boundary conditions of the forward model and the prior distributions, to meet the conditions of the tomography experiment at each site. Moreover, different types and sources of measured data can be processed for the inversion, such as tracer or in-situ stress data, provided that a forward model is available that allows for the flexible update of DFN parameters. The workflow for the setup of the inversion problem is similar. The basis is the properties of the fractures intercepted by the boreholes, i.e., their position and orientation, obtained from optical or acoustic televiewer logs or outcrops. This knowledge is utilized for the prior distributions on the spatial parameters and for the specification of fracture sets. The prior distributions on the hydraulic parameters are based on cross-hole flow tests in this study. This can also be done by the evaluation of the hydraulic tomography experiments as continuous hydraulic conductivity and specific storage tomogram."

*Overall, my recommendation is either to recast the manuscript as a technical note or case study with revisions, or to rework the analysis with more of the available pressure data from some of the other 13 of 15 wells - and/or new tests and observation zones as suggested above - so a potentially more-realistic, alternative fracture network might be revealed and evaluated more clearly (i.e., with less reliance on constraining priors and assumptions).*

Recasting the manuscript as a technical note or as a case study would not change the content of the manuscript. We would leave this to the editor to decide if the presented work cannot

be categorized as a regular manuscript. Reworking as recommended by the reviewer would be appealing, especially, by adding more tests and observation zones. However, as already replied above, this would not only mean starting from scratch again with the entire inversion procedure, but also require a considerable amount of costly field work which is not realistic for us. Moreover, as explained in the manuscript (Line 86–94), the investigated zone is the most permeable and highly fractured part of the site which was investigated by the ISC experiments.

*Specific Comments:*
*L104-105 and L157-164: Why apply noisy data directly rather than use smoothed data (e.g., can be as simple as a moving average) and thereby reduce the error and uncertainty? See L232-234 where the text acknowledges the reduced quality and meaning of prediction uncertainty due, at least in part, to measurement noise.*

> The MCMC algorithm as a stochastic method is designed to interpret the input data in a probabilistic way, in this case, as a normal distribution, whereby the standard deviation of the normal distribution accounts for errors (measurement and model). Smoothed data means a standard deviation of zero making the evaluation of the likelihood function (Eq. 3) impossible (division by zero).
> We added that in Line 105:
> "Due to the stochastic inversion approach, t̶The noisy pressure response data can be applied directly for the inversion without the necessity of smoothing or filtering the signals."
> See also Line 154 to Line 157:
> "The variance $\sigma^2$ in the likelihood function (Eq. 3) accounts for different sources of uncertainties like measurement errors, modeling errors, and errors of the conceptual model. Therefore, the value of the variance is estimated separately for each pressure signal. This is implemented as a part of the inversion algorithm after the update of the parameters of the DFN. The measured data is assumed to consist of a mean and a normally distributed error $\boldsymbol{d} = \bar{\boldsymbol{d}} + \mathcal{N}(0, \sigma^2)$."

*L123-129: S3 shear zones are given impermeable cores but can be traversed by placing fractures across them. This makes sense as a strategy to manage realism with simplicity. Are any such fractures that cross S3 shear zones evident and placed in the modeling results? It is difficult to tell from Fig 6.*

> Based on this question, we understand that our original explanation regarding the shear zones was misleading. Therefore, we revised this in the new version of the manuscript.
> Yes, there are fractures close to S3.1 with a probability of $\approx 40\%$ (visible in Fig. 6, mainly for z = 17 m, closer to injection borehole #2). Therefore, we conclude that either the shear zone is more permeable in this region or that permeable fractures are present close to the boundary of the shear zone. This is supported in the results by identifying a high probability of fractures not by traversing the shear zone, but by placing fractures of fracture set 1 parallel to the shear zone.
> Line 184-196:
> "The investigated volume is limited to the volume between the two S3 shear zones (Fig. 1). The shear zones consist of a fault core and a damage zone. The permeability increases with the distance to the fault core whereby the cores are almost impermeable (Wenning et al., 2018). Since the properties of the shear zones are not the target of this study, the shape is simplified and the associated hydraulic parameters are fixed. The shape of the shear zones is simplified to a plane rectangle, i.e., a linear interpolation between the shear zones' traces at the injection boreholes. A constant hydraulic aperture of $a_{SZ} = 1 \cdot 10^{-5}$ m is assigned. This small value is chosen based on preliminary in-situ tests and the knowledge, that the cores of the shear zones are impermeable at their tunnel intersection. A higher permeability of the shear zone c̶o̶r̶e̶s̶ at specific locations can be covered by placing fractures in the respective area which accounts also for the spatial variability of the permeability of the shear zone c̶o̶r̶e̶s̶. The specific storage value is also fixed to $S_{SZ} = 1 \cdot 10^{-5}$ Pa$^{-1}$. The high value is prescribed d̶u̶e̶ ̶t̶o̶ ̶t̶h̶e̶ ̶c̶e̶n̶t̶i̶m̶e̶t̶e̶r̶-̶w̶i̶d̶e̶ ̶o̶p̶e̶n̶i̶n̶g̶ ̶o̶f̶ ̶t̶h̶e̶ ̶s̶h̶e̶a̶r̶ ̶z̶o̶n̶e̶ ̶a̶n̶d̶ considering the results from crossborehole tests (Klepikova et al., 2020). Fractures of the fracture set 1 are approximately parallel to the S3 faults. Hence, a position close to an S3 fault also accounts for spatial changes in the permeability and specific storage of the S3 faults."

Line 278-282:

"Several fractures of fracture set 1 appear close to the S3.1 shear zone, indicating either permeable fractures close to the shear zone or a higher permeability of the shear zone in this region than the assigned value. This demonstrates that the prescribed assumptions on hydraulic properties of the shear zone do not induce crucial conceptual constraints in the inversion, but a locally high permeability of a shear zone is indicated by a locally high fracture probability."

*L133-139: Boundaries of specified pressure (at AU) and Robin type (at VE), and S3 shear zones with fixed impermeable cores, are so close to the HT test volume that they likely have effects on observed and modeled pressure changes (especially since rock matrix outside of modeled fractures is treated as impermeable). Commonly models move boundaries out far enough to have minimal effects, but that is not possible in this case. Some of the time-buildup curves in Fig 6 seem to under-match the observations suggesting perhaps "buffering" effect with boundaries? Perhaps this could be checked by modeling the HT tests with several very low to intermediate to actual injection rates to see if the build-up curves become affected progressively?*

The outcome of the simulated hydraulic tomography experiments is affected mainly by the AU outflow which corresponds with the real conditions at the site. A pressure or outflow boundary condition was not realistic for the VE boundary since we cannot model all the way to the VE tunnel. Therefore, we chose a transfer boundary condition as suggested by Watanabe et al., 2017. We simulated the HT experiments with a progressively increasing injection rate as suggested by the reviewer for a few posterior DFN realizations and the simulated pressure response curves increased according to the injection rate.

Line 134-136:

"The way to the VE tunnel cannot be modeled explicitly. Therefore, we apply a Robin boundary condition as a transfer boundary condition to consider the transition of the flow and the extension of the shear zones towards the VE tunnel (Watanabe et al., 2017). "

*L169: For reference, since computing cost trade-off for accuracy (or for problem size?) is a stated consideration, it might be good to add what computing and time resources were used for the full set of realization runs. This type of information is commonly included in HT papers on new methods or applications.*

The computation of the inversion was executed by an Intel Core i9 Workstation with 10 Cores and 128 GB RAM and lasted approximately one week. We included this in the revised manuscript.

Line 229-232:

"Overall, 27,000 DFN realizations are considered as posterior DFN realizations since they minimize the error and fulfill the prior conditions. DFN realizations from the initial 500 iterations are discarded as so-called burn-in iterations due to a higher error. The computation of the inversion was executed by an Intel Core i9 Workstation with 10 Cores and 128 GB RAM and lasted about one week."

*L172-177 (and Fig 6): Important fixed modeling simplifications include 2 sets of fractures that have fixed orientations and that are surrounded by impermeable rock matrix. However, apparently contrary to these rules, Fig 6 shows several fractures at distinctly different angles than the 2 sets with fixed orientations - see panels for highly probable fracture locations and for highly probable large aperture occurrences at z=17m and z=16m in Fig 6. This occurrence of apparently disallowed fractures strongly suggests that modeling of all the test data with only the 2 fixed fracture sets does not fully compensate for additional fractures and fracture network structure. That is, it appears that there is a non-*

*uniqueness issue that may be larger than just this Fig 6 non-allowed occurrence, and therefore should be acknowledged (in Section 4.3 and Section 5, L275-276), and should be somehow assessed.*

Fractures that appear with a different orientation, i.e., not with one of the orientations defined for the fracture sets, are the fractures intersected by the injection boreholes. For these fractures, the true orientations as obtained from the optical televiewer logs are assigned. Moreover, the position of these fractures is fixed, therefore, they appear with 100% probability. Estimation of the exact orientation of the fractures is not part of the inversion problem. Instead, the parameters with a major influence on the flow are adjusted by the inversion algorithm. Moreover, note that not one deterministic fracture network is evaluated that results from minimizing an objective function, but a fracture probability that stems mainly from fracture position and length.

Line 257-260:

"The FPM and the mean hydraulic aperture are shown for different cross sections $z$ in Fig. 6. The fractures intercepted by the injection intervals and the shear zones are fixed, and therefore, they appear with a probability of 100%. Their orientation as derived from the optical televiewer logs is assigned to these fractures, therefore, the orientation is in the same range as the orientation defined for the fracture set, but the exact values vary."

Line 239-243:

"The DFN realizations are evaluated by a fracture probability map (FPM) over the investigated volume. For this, the inspected rock volume is divided into raster elements. Each element records if the element is part of a fracture. By taking the mean element-wise over all the posterior DFN realizations, the probability of each raster element for being part of a fracture is derived. The evaluation of the FPM summarizes the estimated position and length of the fractures, i.e., those parameters with major influence on the flow."

*L210: 27,000 DFN realizations are considered as posterior DFN realization. How many total realizations were run?*

In total 27,500 iterations were run. After a large decrease in the overall error and a comparison between simulated and measured pressure signals, the remaining iterations were considered posterior realizations.

Line 229-232:

"Overall, 27,000 DFN realizations are considered as posterior DFN realizations since they minimize the error and fulfill the prior conditions. DFN realizations from the initial 500 iterations are discarded as so-called burn-in iterations due to a higher error. The computation of the inversion was executed by an Intel Core i9 Workstation with 10 Cores and 128 GB RAM and lasted about one week."

*L284-287: The title suggests emphasis on HT, but future research recommendations here and description of other possible applications elsewhere (next comment) only consider more priors and prior treatments rather than more-capable HT designs and execution – such as greater testing and observation coverage to get more-definitive hydrologic evidence on the hydrologic heterogeneity of the main fracture system and its associated nearby fracture network. Cost and time are often raised as obstacles for HT, but it might be good to consider how much additional "direct" HT coverage could be attained for at least some of the cost and time of the many other "indirect" types of data and investigation that also have associated costs and time for collection, analysis, and efforts to relate data to hydrology as priors such as for this paper.*

The decision about acceptance/rejection is based mainly on the likelihood function, i.e., on the comparison of simulated data and measured HT data. The influence of the HT data with respect to the prior on the hydraulic aperture is discussed in the evaluation of the results.

Line 269-271:

"Although a maximum hydraulic aperture of $10^{-3}\ m$ is enabled by the prior distribution, only a few fractures with a small probability appear with an aperture close to the maximum possible value as visible in Fig. 6 at a depth of z = 17 m."

We extended the conclusions of the paper to discuss the necessary steps for transferring the method to other sites with a special emphasis on larger scales (see our replies above and below).

*L290-294: This research at the Grimsel site may be focused on the connectivity of higher conductivity fractures between 2 shear zones, but other applications (e.g., repository leakage, directed circulation for in-situ remediation of contaminated source zones, multi-scale effects in critical zones…) may need quantitative understanding of less spatially restricted fracture occurrence, properties, connectivity, and network flow behavior – even in relatively small investigated volumes similar to this study. Can the method of this paper feasibly be adapted to get a full fracture network realization in a volume of a few 10s of meters in three dimensions, or scaled to larger volumes?*

We explained the necessary steps and input for the application of the method to other sites in the conclusions. We agree that the application on a larger scale requires caution and more research.

Line 320-340:

"The introduced inversion framework can be applied in a highly flexible way for the characterization of different fractured sites by adapting the site-dependent parameters,  to meet the conditions of the tomography experiment at each site. Moreover, different types and sources of measured data can be processed for the inversion, such as tracer or in-situ stress data, provided that a forward model is available that allows for the flexible update of DFN parameters. The workflow for the setup of the inversion problem is similar. The basis is the properties of the fractures intercepted by the boreholes, i.e., their position and orientation, obtained from optical and/or acoustic televiewer logs or outcrops. This knowledge is utilized for the prior distributions on the spatial parameters and for the specification of fracture sets. The prior distributions on the hydraulic parameters are based on cross-hole flow tests in this study. This can also be done by the evaluation of the hydraulic tomography experiments as continuous conductivity and specific storage tomogram. As the definition of priors and constraints delineates the range of feasible DFN realizations, this step has to be done carefully. However, the presented Bayesian framework allows the combination of multiple and diverse hard and soft data, which often exists in addition to hydraulic test data that is used to guide the inversion. As demonstrated here, too tight constraints may be avoided by uniform prior distributions with large value ranges at the expense of higher computational costs for the inversion. In practice, the amount of information describing the fractured rock is determined mainly by the hydraulic tomography data, i.e., by the number of intervals and boreholes.

The present study paves the way towards the applicability of the discrete inversion approach on a larger scale. The main issue will be to balance the degree of field testing with the desired fracture resolution and the associated computational costs. One possible direction is explicitly implementing only large conductive fractures. The role of smaller fractures with a lower permeability could be represented by calibrating a background permeability within the discrete fracture matrix approach (Berre, 2019). Another appealing direction is the representation of scale-dependent fracture sets by their statistical properties following a hierarchical parameterization (Ma et al., 2020)."

*Technical Corrections:*
*Eqn 1 - T subscript for gradient operator needs to be explained.*

We added that in the revised manuscript in Line 116:

"The subscript T of the gradient ($\nabla_T$) denotes that it is evaluated in the fracture plane (Zimmerman and Bodvarsson, 1996, Berre, 2019)".

*Figure 4 - Shows only 2 gray and 1 orange data points for fracture set 2. Is this the full basis of support for this major fracture set? Unclear what the gray dots are vs what the colored (blue and orange) dots are.*

Yes, the orientations of the two fractures are the full basis of support for the second fracture set. It is not a particularly robust choice, but the data shows that there are fractures with this orientation. The existence of a second fracture set with an orientation close to the orientation defined for the second fracture set is also supported by the optical televiewer logs from other boreholes (see FBS1, PRP1, PRP2) (Krietsch et al., 2018). This is a clear indication that the orientation has to be considered. We considered assigning a higher proposal probability for inserting fractures of the first fracture set, however, we decided against it to avoid additional prior restrictions. Instead, the HT data is the main factor to decide about acceptance/rejection in the update probability of the inversion algorithm. We also revised the figure and the caption.

*Figure 5 – Recommend replotting on log-log axis scales to better assess the hydrologic responses in this common analytic frame of reference. Also it is difficult to distinguish between the two gray shades of prediction and posterior 95% limits. Perhaps add one generic axis label for pressure change on the left and one generic axis label for Elapsed time at the bottom, and change column labeling from caption letter codes to names of zones in wells at the top. Be explicit that plots of the injection zones per test are on the diagonal.*

We revised the figure and the caption according to the referee's suggestions, i.e., we included generic labels for the time and the pressure response and used a more distinct contrast for the posterior and prediction limits. However, the present visualization compared to a log-log axis seems better to us and makes our results comparable with other studies characterizing the same region of the Grimsel test site with similar HT data (Klepikova et al., 2020).

*Figure 6 – It is difficult to relate the z positions to the model volume (no z axis or magnitudes shown in the very small cuboid reference diagram). Also it is difficult to read aperture magnitudes with the increment labeling on the shading scale. The text refers to east positioning (L255), so direction labels are needed here, and also in Fig 1. Enlarging the reference diagram would be helpful.*

We revised the figure providing a larger reference diagram. We chose a diverging colormap (instead of a sequential colormap), which should make it easier to evaluate the aperture. We also clarified in the revised manuscript that $x$ refers to easting +667,400 m, $y$ to northing +158,800 m, and $z$ to height +1,700 m (Line 217-218).